# Shortening of Overall Orthodontic Treatment Duration with Low-Intensity Pulsed Ultrasound (LIPUS)

**DOI:** 10.3390/jcm9051303

**Published:** 2020-05-01

**Authors:** Harmanpreet Kaur, Tarek El-Bialy

**Affiliations:** 1Division of Oral Biology, School of Dentistry, Katz Group for Pharmacy and Health Research, University of Alberta, Edmonton, AB T6G 2E1, Canada; kaur3@ualberta.ca; 2Division of Orthodontics, School of Dentistry, Katz Group for Pharmacy and Health Research, University of Alberta, Edmonton, AB T6G 1C9, Canada

**Keywords:** orthodontic tooth movement, non-invasive therapy, low intensity pulsed ultrasound, LIPUS, clear aligners

## Abstract

The aim of this retrospective clinical study was to determine if there is a reduction in the overall treatment duration in orthodontic patients using low-intensity pulsed ultrasound (LIPUS) and Invisalign SmartTrack® clear aligners. Data were collected from the first thirty-four patients (9 males, 25 females; average age 41.37 ± 15.02) who finished their orthodontic treatment using an intraoral LIPUS device and Invisalign clear aligners in a private clinic. The LIPUS parameters used by patients at home for 20 min/day were: ultrasonic frequency 1.5 MHz, pulse duration 200µs, pulse repetition rate 1 kHz, and spatial average-temporal average intensity 30mW/cm^2^. A control group (11 males, 23 females; average age 31.36 ± 14.41) matching for the same malocclusions was randomly selected from finished treatment cases of the same clinician. The date of first Invisalign attachment placement and first use of LIPUS application was recorded as T0, and the date of retainer delivery was recorded as T1. The treatment duration (T1–T0) and treatment reduction percentage with LIPUS device were collected and analyzed using two-sample t-test in Microsoft Excel. Treatment duration was significantly reduced in the LIPUS group (541.44 ± 192.23 days) compared to control group (1061.05 ± 455.64 days) (*p* < 0.05). The LIPUS group showed on average 49% reduction in the overall treatment time as compared to the control group. The average compliance of the patients using LIPUS was 66.02%. Patients who used LIPUS showed a clinically significant reduction in the overall orthodontic treatment duration compared to the control group who used Invisalign clear aligners only.

## 1. Introduction

Malocclusion is defined as misalignment of teeth and/or jaws in any or all the three dimensions of space. It can cause abnormal wear of tooth surfaces, difficulty in speaking and chewing, strain on the supporting alveolar bone and gums, and possible temporomandibular joint dysfunction [1]. Tooth roots are covered by special mineralized tissue cementum that is connected to the alveolar bone through the surrounding highly vascularized soft connective tissue, the periodontal ligament (PDL) [2]. Unlike the physiological tooth movement, orthodontic tooth movement (OTM) is a complex process of bone remodeling that occurs in response to the externally applied mechanical forces through wires and brackets or clear aligners [3]. Different types of bone cells are within the alveolar bone, including osteoblasts, osteoclasts, osteocytes, and bone lining cells [4].

OTM was first described as the “Pressure–Tension” theory by Oppenheim [5] and Schwarz [6]. During the OTM, the side towards which the tooth is moving is the pressure side, while the opposite side is the tension side. The compression of blood vessels in PDL on the pressure side leads to decreased nutrient flow, stenosis, and formation of necrotic tissue [7,8]. This inflammatory process causes migration of phagocytic cells, like macrophages, giant cells, and osteoclasts, which further leads to bone resorption on the pressure side of PDL. Bone resorption at the bone and PDL interface is the rate-limiting factor for OTM [9]. An important factor in the orthodontic practice success is to precisely or approximately estimate the treatment duration. With an increasing number of adult patients seeking orthodontic treatment, where the OTM is known to be slower than in adolescents, the research and innovation in the orthodontic field lead to modification in treatment protocols. For example, change in orthodontic biomechanics by using low friction/frictionless orthodontic techniques, and development of various techniques to accelerate OTM, including pharmacological agents (e.g., parathyroid hormone, Vitamin D3, Prostaglandins) [10], magnetic fields [11], corticotomy [12], distraction osteogenesis [13], low-level laser [14], and mechanical vibration [15].

Low-intensity pulsed ultrasound (LIPUS) is one of the non-invasive, non-pharmacological methods to accelerate OTM that has been used in the medical field for over six decades as in sports medicine, myofunctional therapy, joint stiffness reduction, increase muscle mobility, and healing of non-healing bone fractures [16]. It is a form of acoustic pressure wave which, when it passes through the living tissues, causes micromechanical strain, resulting in cascades of molecular events [17]. In the previous in-vitro, animals, and human studies, LIPUS has shown to minimize orthodontically induced tooth root resorption (OITRR), accelerate orthodontic tooth movement, and increased expression of collagen 1 (Col1), alkaline phosphatase (ALP), osteoprotegerin (OPG), and receptor activator of nuclear factor-kappa β-ligand (RANK-L) [2,18,19,20,21,22,23]. The aim of this retrospective study was to analyze the overall treatment duration and percentage treatment reduction if any in the patients using a commercially available LIPUS system for intraoral use with Invisalign clear aligners and compare these variables with patients who were treated by Invisalign clear aligners only.

## 2. Methods

### 2.1. Study Design

This retrospective clinical study has been approved by the Human Research Ethics Board at the University of Alberta, Canada (Protocol number Pro00032422). The data of the first thirty-four patients (9 males, 25 females; average age 41.37 ± 15.02) who completed their orthodontic treatment with LIPUS intraoral device concurrent to using Invisalign clear aligners in a private clinic was collected and analyzed. The same orthodontist performed all the orthodontic procedures. A control group (11 males, 23 females; average age 31.36 ± 14.41) matching for the same malocclusion to the LIPUS group was randomly selected from the clinic’s finished treatment cases. The following inclusion criteria were applied:Good oral hygieneFull permanent dentitionPatients with no medical historyPatients with no history of medicationNon-pregnant womenPatients undergoing orthodontic treatment with clear aligners only

No other additional criteria were applied while selecting the control group, other than the finished cases with the type of malocclusion. All cases were treated by non-extraction treatment Informed consent was signed by all the patients and/or guardians to use their data for research purposes. All the patients were treated by Invisalign SmartTrack^®^ (Align Technology, Santa Clara, CA, USA) clear aligners programmed at the default aligner rate of tooth movement of 0.25 mm maximum per aligner. All the patients were given instructions on how to place and remove their aligners from the mouth. They were instructed to wear their aligners for 20–22 h per day and to change the aligners as soon as they become loose.

### 2.2. LIPUS Device

LIPUS was applied using the Aevo System (SmileSonica Inc., Edmonton, AB, Canada). The LIPUS parameters were as follows: ultrasonic frequency 1.5 MHz, pulse duration 200µs, pulse repetition rate 1 kHz, and spatial average-temporal average intensity 30mW/cm^2^. It is a non-invasive, battery-powered, portable, and intended to be used for 20 min/day at home. The device consists of three main components (Figure 1).

A:Handheld electronics: It controls LIPUS treatment delivery and provides information regarding treatment procedure and status. It is powered by a rechargeable battery. The information displayed on the screen includes the current status of the device, remaining treatment time, battery charge level, and current date and time. It also maintains a complete record of treatment parameters.B:Mouthpieces: The device has two mouthpieces, one for the mandible arch treatment and the other for the maxilla arch treatment. Each mouthpiece is similar to a mouthguard and consists of 10 ultrasound emitters set inside a flexible biocompatible encapsulation. All the internal components are hermetically sealed to prevent contact with saliva. The mouthpiece is attached to the handheld electronics with a cable.C:Ultrasound coupling gel: A tasteless gel provided in single use pouches is applied to the inner walls of the mouthpiece before the start of each treatment. Patients were instructed to apply a thin layer so that LIPUS can be properly transmitted from the mouthpiece through gums to the alveolar bone surrounding the teeth roots.

### 2.3. Data Collection

General data, such as age, gender, type of malocclusion, and start and finish date of the orthodontic treatment with Invisalign, were collected. The date of first Invisalign attachment placement and first application of LIPUS was recorded as T0, and the date of Invisalign attachment removal was recorded as T1. The treatment duration, i.e., T1–T0, average number of days per tray were collected and analyzed, and overall treatment reduction percentage was calculated.
$$\mathrm{Overall}\;\mathrm{treatment}\;\mathrm{reduction}\;\mathrm{percentage}\;=\;\left\{\frac{\mathrm{Average}\;\mathrm{treatment}\;{\mathrm{days}}_{\mathrm{Control}}\;-\mathrm{Average}\;\mathrm{treatment}\;{\mathrm{days}}_{\mathrm{LIPUS}}}{\mathrm{Average}\;\mathrm{treatment}\;\mathrm{days}\;{}_{\mathrm{Control}}}\right\}\;\times \;100$$

### 2.4. Statistical Analysis

Descriptive statistics (Mean and Standard Deviation) were calculated for all the collected variables in both the groups. Statistical comparison with Student’s t-test for independent samples were performed on T1–T0 treatment duration and on the average number of days per tray. All the statistical analyses were performed using Microsoft Excel 2016, with *p*-value less than 0.05 being considered significant.

## 3. Results

### 3.1. Subjects

Subjects: From the thirty-four patients group treated with LIPUS device, there were 9 males and 25 females. The average age of the LIPUS treated group was 41.37 ± 15.02 (minimum 16 years 4 months and maximum of 72 years). In the control group, there were 11 males and 23 females. The average age of the control group was 31.36 ± 14.41 (minimum 15 years and maximum 64 years and 6 months).

The number of patients in each class of malocclusion is presented in Table 1.

### 3.2. Number of Days per Tray

Patients treated with LIPUS device (6.02 ± 1.49) showed a significant difference in the number of days per tray worn as compared to the control group (10.81 ± 3.31) (*p* < 0.05) (Figure 2). Figure 3 depicts the number of days per tray for each malocclusion, and the difference was statistically significant in each type of malocclusion.

### 3.3. Treatment Duration

The treatment duration was significantly reduced in the LIPUS treated patients (541.44 ± 192.23 days) as compared to the control group (1061.05 ± 455.64 days) (*p* < 0.05) (Figure 4), and the difference was statistically significant in each malocclusion (*p* < 0.05) (Figure 5).

All in all, the patients treated with LIPUS during their orthodontic treatment with Invisalign showed a 49% reduction in the overall treatment time as compared to the patients undergoing orthodontic treatment with Invisalign alone. The patient average compliance using the Aevo System was 66.02% according to the internal microchip built into the Aevo System that records every time the patient uses Aevo System. [23].

## 4. Discussion

With an increase in the number of adult patients for orthodontic treatment and interest in accelerating tooth movement to shorten the treatment duration, many technologies have been developed and many are still in the research and development phase.

LIPUS is one such form of non-invasive technology that has been used in the medical field for over six decades. In the dental field, it has demonstrated significant acceleration of orthodontic tooth movement and reduction of orthodontically induced tooth root resorption in both animal and human studies [21,22,24]. In a prospective multi-center randomized controlled clinical trial [23], the rate of tooth movement increased on average by 29%. The current retrospective clinical study analyzed the effect of LIPUS on the orthodontic treatment time reduction using Invisalign clear aligners. The results showed that patients using LIPUS system during orthodontic treatment were able to shorten the overall treatment duration on average by 49% as compared with the control group, while the average compliance using the LIPUS system was 66.02%. The difference between the two studies could be explained by that in the multicenter clinical trial [23], orthodontic treatment was performed using fixed orthodontic appliances by 5 different clinicians; however, in the current study, treatment was performed by one orthodontist only using clear aligners.

OTM is a bone remodeling process in which there is an interplay of different cell types, such as osteoblasts, osteoclasts, and osteocytes. Bone resorption, caused by activation of osteoclast, is regulated by tumor necrosis factor (TNF) receptor-ligand family which includes OPG, receptor activator of nuclear factor kappa-β (RANK), and RANK-ligand (RANK-L). During mechanical stress application in the form of orthodontic force, the osteocytes release RANK-L, which binds with RANK, stimulating pre-osteoblast fusion, osteoclast differentiation, proliferation, and survival [25,26,27]. OPG is a soluble decoy receptor that prevents the binding of RANK-L to RANK, hence inhibiting osteoclast formation [28]. RANK-L and OPG are important in regulating bone remodeling during tooth movement [29]. Furthermore, vascular endothelial growth factor (VEGF) is increased during OTM, which prevents apoptosis of osteoblasts, stimulates osteoprogenitor cell recruitment, and promotes mineralized nodule formation and release of ALP [30,31]. Several factors affect the OTM, including magnitude of orthodontic force, type of tooth movement, and general and periodontal health of the patient [32,33]. With an increasing number of patients from all age groups, orthodontists need to look at different treatment modalities for more efficient and safer treatment, in addition to applying lower forces for OTM [34]. The accelerated rate of tooth movement in this study could be due to the fact that LIPUS induces strain affecting mechanosensitive receptors, such as integrins, stretch-activated channels on the cell membrane [35]. These receptors further initiate the cascade of cellular and molecular events in the cell known as mechanotransduction. Several cellular signaling pathways, like focal adhesion kinase (FAK) [36], mitogen-activated protein kinase (MAPK) [37], and Rho pathways [38], have shown to be activated in the in-vitro studies with LIPUS application. Through these mechanisms, LIPUS has shown to enhance bone formation and osteoblast differentiation in fracture healing cases. It has also shown to promote angiogenesis by upregulating VEGF expression in human osteoblasts [39], in wound healing [40], early osteogenesis by upregulating insulin-like growth factor which mediates osterix expression [41,42], and increased expression of osteogenic markers, i.e., collagen I [43], osteocalcin [44], osteopontin, and bone sialoprotein [45,46]. LIPUS increases the proliferation of osteoprogenitor cells with increased expression of bone morphogenetic protein 2 (BMP-2), BMP-7, and runt-related transcription factor (Runx2) [39,47,48]. Runx2 is a transcription factor for osteoblast differentiation from mesenchymal stem cells. A study by Xue et al. [24] showed increased alveolar bone remodeling by increasing expression of Runx2 and BMP-2 in rat orthodontic model, hence increasing OTM velocity.

It seems that the mechanism of acceleration of OTM by LIPUS has similarities to other accelerating techniques, such as laser, high frequency vibration, and corticotomy, that work through the RANK-RANKL pathway. This may warrant further investigation to compare all techniques in this regard.

Although LIPUS has proved to increase osteogenic markers expression in many studies, it has also shown to regulate osteoclast differentiation through OPG/RANK-L expression. LIPUS at the intensity of 100 and 150 mW/cm^2^ showed a decrease in osteoclast number and activity, and an increase in OPG/RANK-L expression in rats treated with LIPUS [49]. In another study [50], RANK-L gene expression was most profound during the third week of LIPUS application; on the other hand, OPG expression remained constant throughout three weeks in murine osteoblast cell culture. This implies that LIPUS enhances osteoclastogenesis during bone regeneration. In a study by Feres et al. [51], LIPUS showed increase osteoclasts activity in the absence of osteoblasts. These findings support the result of our current retrospective study that on the compression side of orthodontic force application, LIPUS enhances osteoclastic activity while on the tension side, LIPUS accelerates the osteoblastic activity and enhanced bone regeneration, hence accelerating tooth movement and it is safe [52,53]

Another advantage of using LIPUS during orthodontic treatment is the preventive effect on root resorption. Although in the current study we did not analyze the effect of LIPUS on root resorption, previous clinical studies [21,22,23], however, showed a decrease of orthodontically induced tooth root resorption using the same LIPUS treatment parameters.

The current retrospective study is overcoming few of the limitations that were encountered in a previous clinical trial [23], specifically the small patient number included in the split mouth design (21 data pairs from 21 split-mouth patients) and the fact that the effect of LIPUS was only studied during gap closure in Class II malocclusion patients requiring first premolar extraction. The current study extends the knowledge to all classes of malocclusions, uses a larger number of subjects (34 active and 34 control patients), and looks at the overall treatment duration.

## 5. Conclusions

In the current study, patients treated with LIPUS treatment showed faster tooth movement and reduction in the overall treatment time on average by 49%, while the average compliance using the LIPUS device was 66.02%. This study demonstrated the use of LIPUS through the Aevo System during orthodontic treatment using clear aligners significantly reduced the overall treatment duration. 

## Figures and Tables

**Figure 1 jcm-09-01303-f001:**
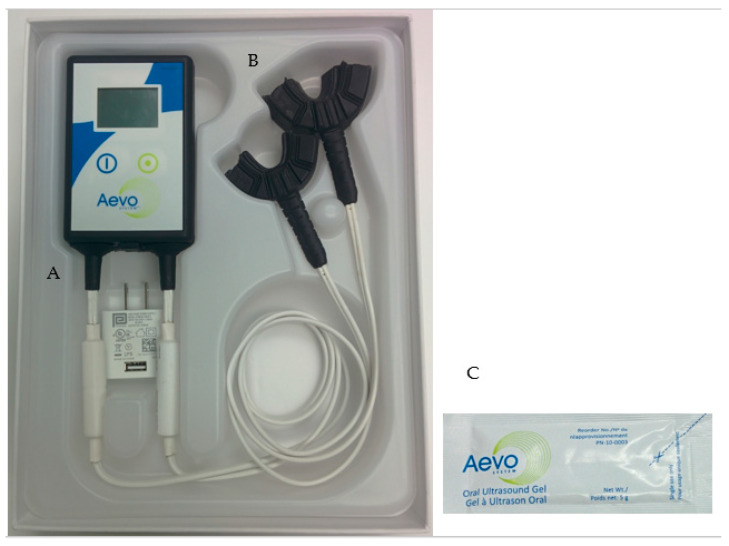
Low-intensity pulsed ultrasound (LIPUS) device (Aevo System). (**A**): Handheld electronics; (**B**): mouthpieces; and (**C**): ultrasound coupling gel.

**Figure 2 jcm-09-01303-f002:**
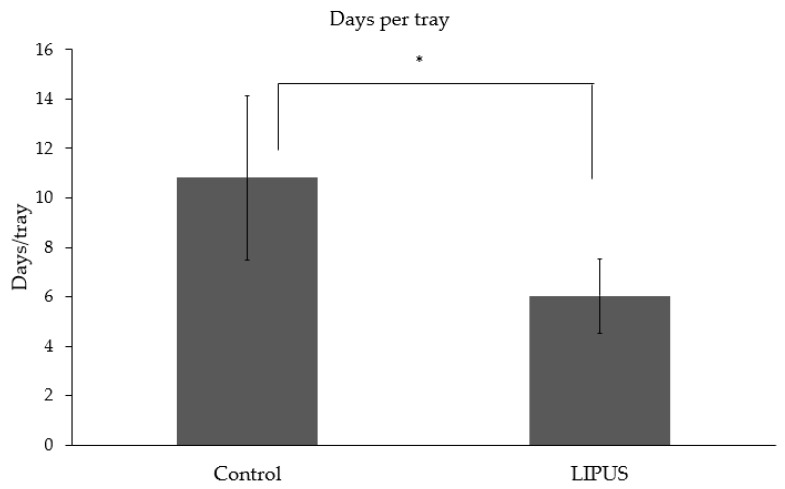
Average number of days per tray/aligner in the control and LIPUS treated group (**p* < 0.05).

**Figure 3 jcm-09-01303-f003:**
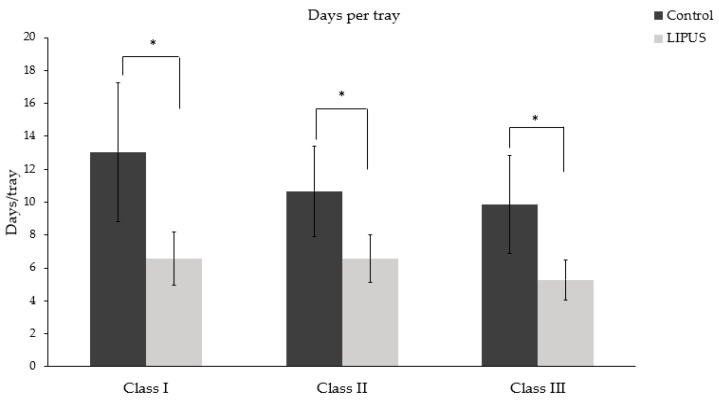
Average number of days per tray in Class I, Class II, and Class III malocclusion (**p* < 0.05).

**Figure 4 jcm-09-01303-f004:**
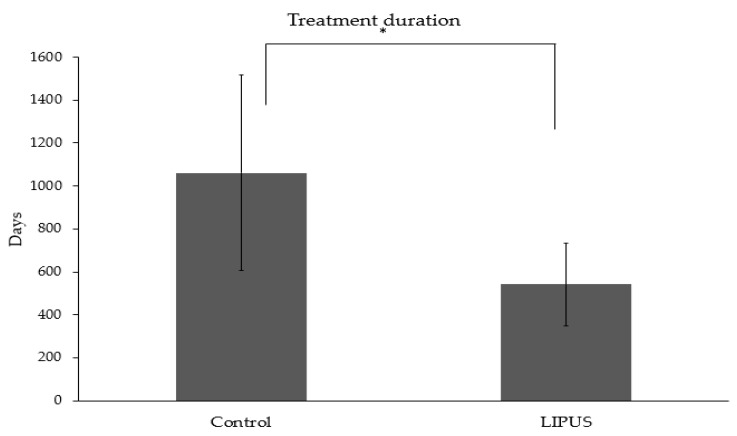
Average treatment duration in the control and LIPUS group (**p* < 0.05).

**Figure 5 jcm-09-01303-f005:**
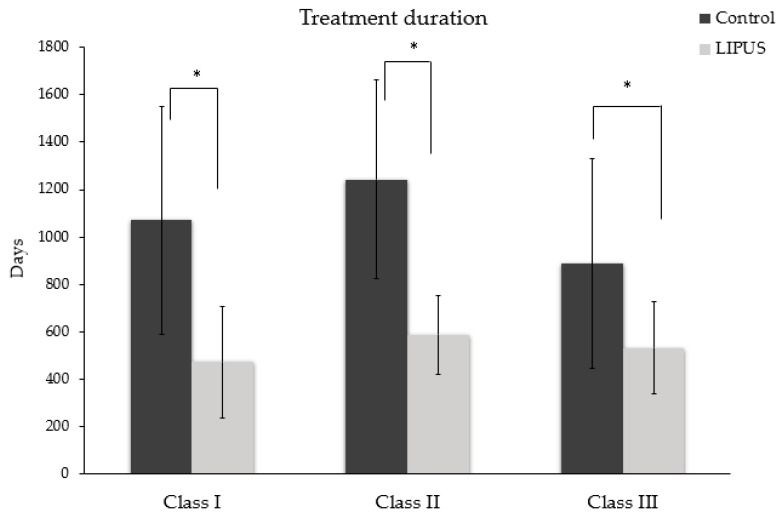
Average treatment duration in Class I, Class II and Class III malocclusion (**p* < 0.05)**.**

**Table 1 jcm-09-01303-t001:** The number of patients in each class of malocclusion.

	Control	LIPUS
Class I	7	7
Class II	13	13
Class III	14	14

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
