# Peer review of "Shortening of Overall Orthodontic Treatment Duration with Low-Intensity Pulsed Ultrasound (LIPUS)"

_jcm, 2020, doi:10.3390/jcm9051303_

Round 1

Reviewer 1 Report

The aim of this retrospective clinical study was to determine if there is a reduction in the

overall treatment duration in orthodontic patients using low-intensity pulsed ultrasound (LIPUS) and clear aligners.

Overall, the article is well written and has clinical implications for the clinicians.

However, I have some minor drawbacks, detailed as follows:

  1. Explain which clear aligners have been used in the abstract?
  2. In the abstract, suggest change the following sentence, line 19  (The variables like treatment duration (T1-T0) and treatment reduction percentage) to (The treatment duration (T1-T0) and treatment reduction percentage …)
  3. Spell out (MS) in (MS Excel)
  4. Please provide a reference to the compliance percentage and how it was chosen.
  5. Please provide additional information about the matching of the study and control groups with regard to: 1. the severity of initial crowding, 2. the type of orthodontic treatment (extraction/nonextraction)
  6. In the conclusion, I would like to see a comparison between LIPUS and other methods for tooth movement acceleration, such as laser, vibrations, corticotomy which all work through the RANK-RANKL pathway.

Author Response

Responses to Reviewer 1 comments

  1. Explain which clear aligners have been used in the abstract?
  2. In the abstract, suggest change the following sentence, line 19  (The variables like treatment duration (T1-T0) and treatment reduction percentage) to (The treatment duration (T1-T0) and treatment reduction percentage …)
  3. Spell out (MS) in (MS Excel)
  4. Please provide a reference to the compliance percentage and how it was chosen.

Compliance was calculated according to the internal microchip built into the AEVO system that records every time the patient uses AEVO system. [23]. (page 2-lines 153-154)  

  1. Please provide additional information about the matching of the study and control groups with regard to: 1. the severity of initial crowding, 2. the type of orthodontic treatment (extraction/nonextraction)

No other additional criteria were applied while selecting the control group other than the finished cases with the type of malocclusion. All cases were treated by non-extraction treatment. Page 2 lines 83-84.

  1. In the conclusion, I would like to see a comparison between LIPUS and other methods for tooth movement acceleration, such as laser, vibrations, corticotomy which all work through the RANK-RANKL pathway.

It seems that the same mechanism of acceleration of tooth movement by LIPUS is similar to other accelerating techniques such as laser, high frequency vibration, ocritcotomy that all work through the RANK-RANKL pathway. This may warrant further investigation to compare all techniques in this regards. Page 8- lines 231-232.

Reviewer 2 Report

i think that the research is well done.

This article of clinical trial has a good methodology even conducted in a small sample size.

The results also confirmed the photobiomodulation effects of ultrasounds on orthodontic tooth movement. 

The amount of patients is low but data are significant

Ultrasounds are a good option in order to reduce orthodontic treatment time. 

Anyway i think it is difficult to find patients able to use this device 20 minutes a day for all the therapy's time.

Author Response

Responses to Reviewer 2 comments

i think that the research is well done.

Thank you

This article of clinical trial has a good methodology even conducted in a small sample size.

Thank you

The results also confirmed the photobiomodulation effects of ultrasounds on orthodontic tooth movement. 

Much appreciated

The amount of patients is low but data are significant

Thank you

Ultrasounds are a good option in order to reduce orthodontic treatment time. 

Thank you

Anyway i think it is difficult to find patients able to use this device 20 minutes a day for all the therapy's time.

Thank you for the comments; this might have been the cause of the compliance of these patients was 66.02%